# Impact of the latent space on the ability of GANs to fit the distribution

## Abstract

The goal of generative models is to model the underlying data distribution of a sample based dataset. Our intuition is that an accurate model should in principle also include the sample based dataset as part of its induced probability distribution. To investigate this, we look at fully trained generative models using the Generative Adversarial Networks (GAN) framework and analyze the resulting generator on its ability to memorize the dataset. Further, we show that the size of the initial latent space is paramount to allow for an accurate reconstruction of the training data. This gives us a link to compression theory, where Autoencoders (AE) are used to lower bound the reconstruction capabilities of our generative model. Here, we observe similar results to the perception-distortion tradeoff (Blau & Michaeli (2018)). Given a small latent space, the AE produces low quality and the GAN produces high quality outputs from a perceptual viewpoint. In contrast, the distortion error is smaller for the AE. By increasing the dimensionality of the latent space the distortion decreases for both models, but the perceptual quality only increases for the AE.

## 1 Introduction

Generative Adversarial Networks (GANs) were introduced by Goodfellow et al. (2014) for the purpose of generative modelling. Since then this framework has been successfully applied to works in style transfer by Karras et al. (2018), superresolution by Shocher et al. (2018) and semi-supervised learning by Salimans et al. (2016), but what GANs actually learn is still poorly understood as has been noted by Webster et al. (2019). Recently, GANs have been used to solve inverse problems, where it was tried to use the generated manifold to solve an auxiliary task like image completion (Webster et al. (2019)), MRI reconstruction (Narnhofer et al. (2019)) or anomaly detection (Shocher et al. (2018)). For those applications, it is necessary to know if the generator NN actually describes the distribution well. Related works have shown that faithfully reconstructing the images from a generator network is not trivial (Webster et al. (2019); Shmelkov et al. (2018)).

The original convergence proof by Goodfellow et al. (2014) assumes that the generator and discriminator Neural Networks (NNs) have infinite capacity and they showed that the discriminator network models the Jensen-Shannon divergence between the probability distribution induced by the generator and the real data distribution. Others have adapted this paradigm and devised loss functions which have been shown to converge to other divergences or distances on the underlying probability distributions (Arjovsky et al. (2017); Nowozin et al. (2016); Mao et al. (2017)). Regularization techniques like Gradient Penalty (Gulrajani et al. (2017)) and Spectral Norm (Miyato et al. (2018)) did improve the stability of the training process (Kurach et al. (2018)) but it is still unclear how well this NNs actually approximate such distances even for trivial problems (Pinetz et al. (2018)). Additionally, it is not at all clear how the generated distribution or the actual target distribution look like. Arora & Zhang (2017) used the birthday paradox to empirically gauge the size of the support of the generated distribution.

GANs are used to transform a well understood low dimensional distribution (in practice either gaussian or uniform) to a high dimensional unknown distribution (Goodfellow et al. (2014)) by playing a min-max game between two NNs. This paper is based around the intuition, that an estimated probability distribution from a dataset $\mathcal{X}$ has high precision if a high percentage of the actual data samples are included in the estimated distribution. To have a sense of what an adequate capacity for

a generator network is, we use AE to reconstruct the dataset first. This work relies on the assumption that it is easier to reconstruct the data samples alone, than to reconstruct the entire manifold and Section 5 shows empirical evidence for this. Based on our intuition the manifold consists of the data samples and imposes additional structure between the data samples. In contrast by just reproducing the data samples, no such additional restrictions are given, making the problem strictly simpler. AEs can be trained rapidly and have been researched in detail for a long time (Rumelhart et al. (1985)). In contrast, trying to do a hyperparameter search on the GAN networks themselves gives rise to all kinds of problems, like instabilities in the training process, random failures and dependence on random seeds for their performance (Lucic et al. (2018)).

Hence, our contributions are as follows:

- An investigation of the impact of the dimensionality of the latent space on the generated manifold. We showcase that the fit of the data depends heavily on the latent space. We also show similar results thereby to the perception-distortion tradeoff(Blau & Michaeli (2018)), where with a small dimension for the latent space, the GAN optimizes for perception and the AE optimizes for distortion.
- Relating the GAN problem to a compression task and furthermore using compression tools via deep learning to produce a lower bound for a dataset dependent suitable dimensionality of the latent space.
- An investigation of the generated manifold and the limitations thereof to produce shifted or noisy images and how this relates to the size of the latent space and overfitting of the generative model.

The remainder of this paper is organized as follows. Section 2 shows the related work. Then in Section 3 we revisit the theory behind pseudo inverting NNs and we explain our methodology in Section 4. In Section 5 the results are shown. Section 6 draws conclusions based on those results.

## 2 RELATED WORK

GANs have been invented by Goodfellow et al. (2014) for the task of image synthesis and semi-supervised learning. The initial formulation has been shown to suffer from mode collapse and vanishing gradient problems and multiple other loss functions have been proposed (Nowozin et al. (2016); Mao et al. (2017); Arjovsky et al. (2017). We choose the Wasserstein GAN, due to it using the L2-distance as its transport cost function to judge the quality of the generated manifold, which is similar to the Mean Squared Error (MSE) used in the classical AE (Rumelhart et al. (1985)). To satisfy the Lipschitz constraint we use Gradient Penalty Gulrajani et al. (2017), due to its good performance in review works(Lucic et al. (2018); Kurach et al. (2018)).

For evaluation purposes we need a quality metrics for GAN and the most common ones rely on the usage of classification models, which are trained on ImageNet (Deng et al. (2009)). Two commonly used metrics are the Inception Score (Salimans et al. (2016)) and the Fréchet Inception Distance (FID)(Heusel et al. (2017)). To combat the bias of having trained on ImageNet, Shmelkov et al. (2018) proposed to train a new classifier to evaluate the GAN model, but this is time consuming and prone to errors by not having a standardized metric. We opted for the FID score, due to its prevalence in related works(Lucic et al. (2018); Kurach et al. (2018)).

AEs have been studied extensively in the past and initially have been proposed by Rumelhart et al. (1985). They have found usage in compression literature (Cheng et al. (2018a;b); Li et al. (2019)). Their first usage in a GAN framework has been shown in the Energy based GAN by Zhao et al. (2016). Since then they have been used as preprocessing to reduce the dimensionality of the data (Guo et al. (2019)), as regularizers (Che et al. (2016)) or as generative models themselves (Webster et al. (2019)).

The inversion of generator networks has been initially used by Metz et al. (2016) and there have been multiple works on how to improve their method (Creswell & Bharath (2018); Lipton & Tripathi (2017)) by adding heuristics like stochastic clipping on top of that. We opted against stochastic clipping, because we already project back to the likely set of candidates. Stochastic clipping would add even more randomness to the results. Generalization of GANs through the detection of overfitting has been proposed recently by Webster et al. (2019). Therein the authors are mostly concerned

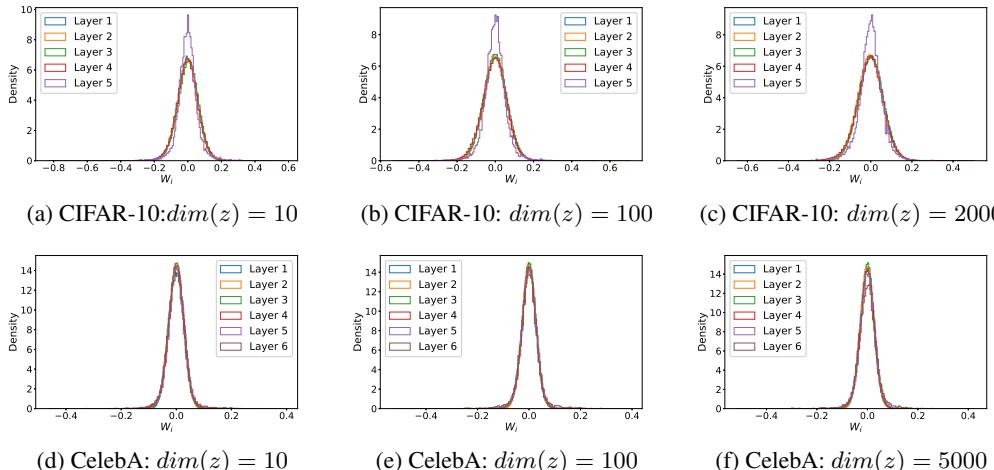

Figure 1: Histogram of the values in the weight matrix and the convolutional layers of a trained WGAN-GP with a latent space specific dimensionality for CIFAR-10 and CelebA datasets. Notice, that the weights share the same characteristics as if they were drawn from a normal distribution.

with detecting memorisation of the training set in the manifold and avoid said memorisation to generalize. In contrast we make the assumption that an estimated probability distribution based on samples should include those samples and overfitting is detected based on a hold out set as well as the ability to reconstruct that hold out set as well as the training set. Additionally, we show that the ability of the GAN to memorize depends mostly on the dimensionality of the latent space and the capacity of the generator network and can be estimated using compression techniques.

## 3 PRELIMINARIES: WHEN ARE NNS INVERTIBLE?

In this work, we want to show the fit of the distribution of trained GAN newtworks using inversions of such networks. This motivates the question of "When are NNs invertible?". Due to the non-convex and non-bijective nature of NNs using ReLU activations invertibility is not given in the general case. The function induced by the NN is in general neither injective nor surjective and it is trivial to construct such cases Arora et al. (2015).

However, if the weights are gaussian distributed, proofs exist, which guarantee that the NN is invertible with a high probability. For this, we revisit the inversion theory of NNs largely pioneered by Arora et al. (2015). They empirically showed that the weights of trained NNs actually exhibit random like properties, enabling the usage of Random Matrix Theory (RMT) to investigate such networks. Theorem 1 in (Ma et al. (2018)) proves that we can invert 2-layer NNs with high probability. In their work they empirically showed that this also holds for multi layer networks. In Figure 1 we show that the same holds true for trained GAN networks. Note, that this property holds across datasets and latent spaces used to train those networks. Glorot initialization (Glorot & Bengio (2010)) was used to initialize these networks. Therefore, the standard deviation used for the initialization process differs for each layer. However, as is demonstrated in Fig. 1, all layers converged to a similar gaussian distribution.

Another indicator, that the optimization process is likely to succeed is given by a trick proposed by Li et al. (2018) for classification networks. It allows us to visualize the loss surface of the optimization problem for one sample at a time. For a particular data point $z^*$, we chose two orthogonal vectors $\alpha, \beta$ and plot the following 2D function:

$$f(a,b) = l(z^0 + a\alpha + b\beta) \qquad (1)$$

We chose $\alpha = z^* - z_0$ to model the optimization path towards the original data point. For this experiment we do not need to run an optimization, because the point is to show that it is possible

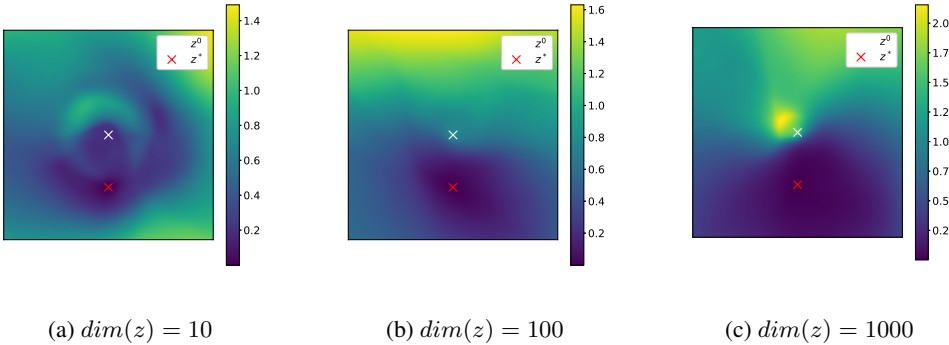

(a) $dim(z) = 10$       (b) $dim(z) = 100$       (c) $dim(z) = 1000$

Figure 2: Loss surface around a data point of the generator of a WGAN-GP trained on CelebA. There is a clear optimization path from $z^0 = \mathbf{0}$ to a point $z^*$ even in this simplified plot, empirically validating the claims of Lipton & Tripathi (2017), that an accurate reconstruction of the generator manifold is possible works using first order methods.

to recover points from the manifold. To ensure that $\beta$ is an orthogonal vector, we simply chose this vector by setting $\beta = \mathbf{0}$ and then $\beta[0] = \alpha[1]$ and $\beta[1] = -\alpha[0]$. Then $\alpha, \beta$ are orthogonal and we can use them as a basis for the plots shown in Fig. 1. For visualization purposes, we scaled $\beta$ to have the same norm as $\alpha$; e.g. $||\beta||_2 = ||\alpha||_2$. We vary $a, b \in [-2, 2]$. In our experiments this always resulted in benign structures as is shown in Fig. 2. If it is trivial to recover points from the manifold and impossible to recover the training images, than this implies that the manifold does not include the training images.

## 4   ESTIMATING A GOOD GAN MODEL FOR A DATASET

Currently, designing new GAN networks works by running a huge hyperparameter search on a plausible set of combinations (Lucic et al. (2018); Kurach et al. (2018)). Lucic et al. (2018) noted that the progress in GANs has actually been quite minimal and vastly overstated, and depends on the hyperparameters more than on the improvements in the methodology, while Kurach et al. (2018) did notice that non-saturating loss in combiniation with spectral norm produced consistently good results, but never the best ones.

### 4.1   LOWER BOUND ON THE GENERATOR NETWORK

The goal of the GAN framework is to encapsulate the high dimensional image distribution in a lower dimensional noise distribution, whereas a generator network maps the lower dimensional distribution to the former high dimensional image distribution (Goodfellow et al. (2014)). This implies that we compress the larger distribution onto the simple lower dimensional distribution and use the generator network as an approximation for the decompression. We rely on the assumption, that less capacity is needed to reconstruct the training set, than to reconstruct the entire manifold. Therefore, to gain some insight into the necessary capacity of the generator network, the usage of an AE is proposed to gauge the number of parameters and the size of the latent space to enable suitable approximation of the training set. In theory, a latent dimension of 1 would be sufficient to reconstruct the entire dataset, by mapping a curve to the datapoints. A simple smooth function that does this for a dataset $\{x_1, .., x_n\} = \mathcal{X} \in \mathbb{R}^{n \times d}$ and $\{p_1, ..., p_n\} = P \in \mathbb{R}^{n \times 1}$ looks like this:

$$g(z) = \sum_{i=1}^{n} x_i \prod_{j=1 \wedge j \neq i}^{n} \frac{(p_j - z)}{(p_j - p_i)} \qquad (2)$$

This function will output the corresponding image $x_i$ for every $p_i$. In practice, this behavior is not observed. AEs with small latent spaces tend to output blurry approximations of the training images as is shown in Section 5. Once a suitable AE for a dataset is found, the decoder part of the AE

is used as the generative model of choice. The AE is trained using the L2-loss, which also is the transport cost function used in WGANs (Arjovsky et al. (2017)) to gauge the quality of the generated manifold. Therefore, the objective function of the AE using a decoder $d$ and an encoder $e$ is defined as follows:

$$L = \min_{d,e} ||d(e(x)) - x||_2 \tag{3}$$

We do not opt for any regularization, because we want to fit the data as closely as possible to determine if the generator capacity is adequate for reconstructing the dataset. In this work we show empirically in Sec. 5 that if the AE already fails to produce crisp results, then also the GAN algorithm fails at having the dataset included in the generated manifold. One thing to note is that, while the encoder part is used as our discriminator, it has a completely different objectives and it is unclear, if a better encoder is also a better discriminator.

### 4.2 Determining the fit of the data in GANs

The key part of this work is to analyse the ability of the GAN to memorise the training dataset, following the assumption, that a probablity distribution based on a sample dataset has to include that dataset. To our best knowledge there is no analytical way to invert generative NNs. Therefore this ill posed problem is stated as an inverse optimzation problem for a function $g$:

$$z^* = \arg\min_z ||g(z) - x||_2^2 \tag{4}$$

This is solved using a standard first order gradient method. However, as is shown in Section 5 the resulting latent vectors are unlikely to stem from the targeted noise distribution, if there are no constraints added. Therefore, the search space for $z$ is restricted to the space of plausible latent vectors. Plausibility in this context means that the probability of the latent vector originating from this space is larger than $99.7\%$ (3-sigma-rule). In the standard multivariate gaussian distribution the norm of the vector $z$ corresponds inversely to the likelihood of $z \in N(0, I)$. Therefore we restrict the norm to be $||z|| \leq \mathbb{E}[||z||] + 3\sqrt{VAR[||z||]}$. For the multivariate gaussian distribution this is easy to calculate and the final objective function is given as follows:

$$z^* = \arg\min_z ||g(z) - x||_2^2$$
$$s.t. \quad ||z||_2^2 \leq dim(z) + 3\sqrt{2dim(z)} \tag{5}$$

This is solved using a projected ADAM optimizer (Kingma & Ba (2014)).

## 5 Experiments

Our experiments are done on medium sized datasets (CIFAR-10 and CelebA ($64 \times 64$ resolution)) and the proof of concept on MNIST is relegated to the supplementary material. The NN adapted for this task is the standard DCGAN from Salimans et al. (2018). Convolutional layers and resize operations are used instead of transposed convolutions, to avoid checkerboard artifacts (Odena et al. (2016)). The full hyperparameter setting is shown in the supplementary material. The visual quality measured by the FID (Heusel et al. (2017)) of the resulting samples is on par with the survey work by Lucic et al. (2018). This work is not aimed at improving the visual quality of the samples generated by GAN algorithms. For CIFAR-10 the test set is used as our validation set and for CelebA we randomly split off $20\%$ of the dataset to be used as validation set.

### 5.1 Relationship between AE and GAN Reconstruction Loss

In principle, getting from an AE to a GAN is just a rearrangement of the NNs. For the GAN training, the default hyperparameter are used as in Gulrajani et al. (2017). The AE NNs are optimized using the ADAM(Kingma & Ba (2014)) optimizer with default hyperparameters for all the datasets,

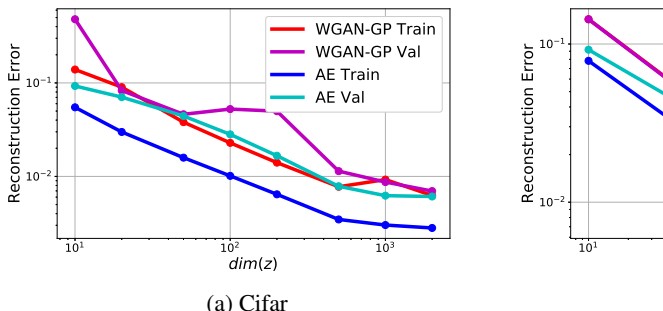

(a) Cifar                              (b) CelebA

Figure 3: Impact of the latent space on the AE and GAN reconstructions loss on CIFAR-10 and CelebA. The AE acts as a lower bound for the GAN reconstruction loss and the reconstruction loss decreases for both the AE and the GAN generator with increasing latent space dimensions.

always on a single GPU. A comparison of the resulting reconstruction errors is shown in Fig. 3. Therein the different AE and GAN models only differ in their latent space sizes. Increasing the latent space dimension also increases the number of parameters. However, for the sake of simplicity this was neglected. The AE network acts as a lower bound for the GAN algorithm, therefore validating our intuition that the AE complexity lower-bounds the GAN complexity.

## 5.2 VISUAL QUALITY OF GANS VS AE GENERATED IMAGES

The visual quality of the generated images is evaluated using the FID (Heusel et al. (2017)) versus the reconstructed images by an AE for the same latent space sizes. The experimental results are shown in Fig. 4 for the CIFAR and CelebA datasets and demonstrate, that the visual quality of the GAN generated images barely changes with different latent dimensionality. In contrast, the AE reconstructions gain quality and sharpness as the latent space increases. This reflects the output of those models shown in Fig. 5, where the resulting images are clearly an approximation of the actual training images. On the other hand, the GAN reconstructions have similar visual quality independent of the actual latent space. However, the resulting images are only remotely similar to the actual image for insufficiently large latent spaces. This phenomenon has been observed in other works, where a similar looking person to an actual training image is found by the inversion procedure (Webster et al. (2019)). However, we show in Fig. 5, that this phenomenon depends mostly on the size of the latent space.

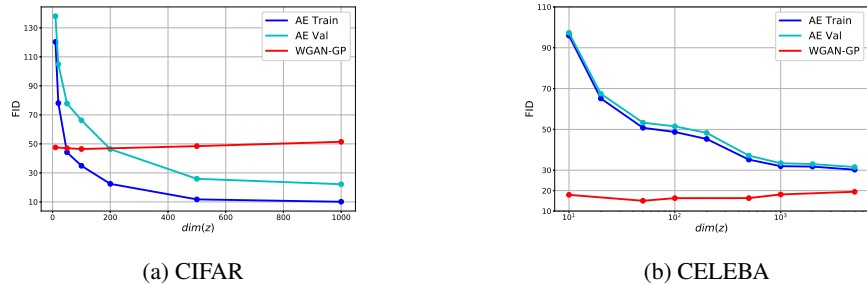

(a) CIFAR                              (b) CELEBA

Figure 4: FID (Heusel et al. (2017)) curve for different latent spaces (lower is better). Increasing or decreasing the latent space does not result in a noticeable difference in the quality metric for the WGAN-GP, but it does for the AE.

## 5.3 WHAT IS INCLUDED IN THE GENERATED MANIFOLD?

While it is possible to reconstruct the dataset accurately using a large enough latent space, in this section we look at what else those models can reconstruct. In these experiments, our baseline model

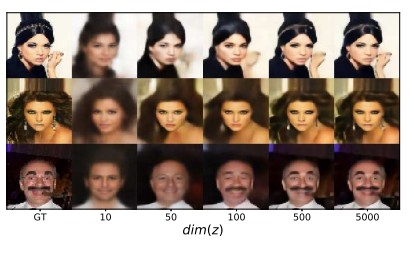

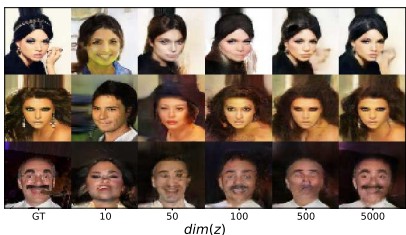

(a) AE

(b) WGAN-GP

Figure 5: The first columns show the original images. As we increase the latent space of the autoencoder the image quality increasing. The GAN reconstructions are of high perceptual quality even for low dimensional spaces, but correspond to other people as has been observed by Webster et al. (2019).

is a WGAN-GP trained on CelebA. The first experiment shown in Fig. 6 uses translated training images. This is especially challenging for this dataset, due to the pre-aligned face images used in the CelebA dataset. As is shown in Fig. 6 the error goes up and the visual quality of the reconstructions deteriorates even for small pixel shifts as long as the latent space is small enough.

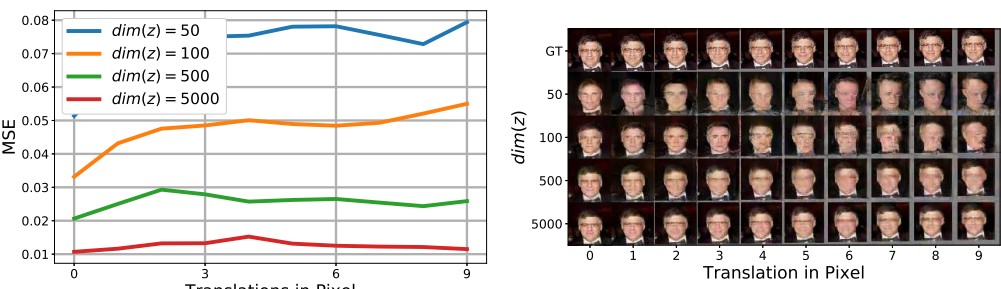

Figure 6: Impact of translation on the reconstruction ability of GAN networks. Notice, that by increasing latent space dimension, the generator network is able to reconstruct translations with a similar precision as the original images.

The second experiment is on reconstructing the CIFAR-10 dataset using the CelebA trained model. To provide the same resolution, the CIFAR-10 images were upscaled to $64 \times 64$ using bilinear interpolation. If the latent space is large enough, we can reconstruct the images well, as shown in Fig. 7. The reconstruction of the CelebA images is always worse as measured by the MSE (Fig. 7).

The final experiment is on reconstructing noise images. For this a dataset of uniform noise images with size $64 \times 64$ is used. The results are shown in Fig. 8. As is suggested by the results, it is in fact not possible to reconstruct everything. One thing to note though is that with lower latent spaces some face structure is visible in the reconstructed images, showing that those models have a strong prior on the face structure. We noticed and the corresponding experiment is shown in the supplementary material, that the norm of the latent vector is smaller than of random vectors. Our hypothesis is that those images are the base face features, which are refined with larger latent vectors, but this guess needs further validation work.

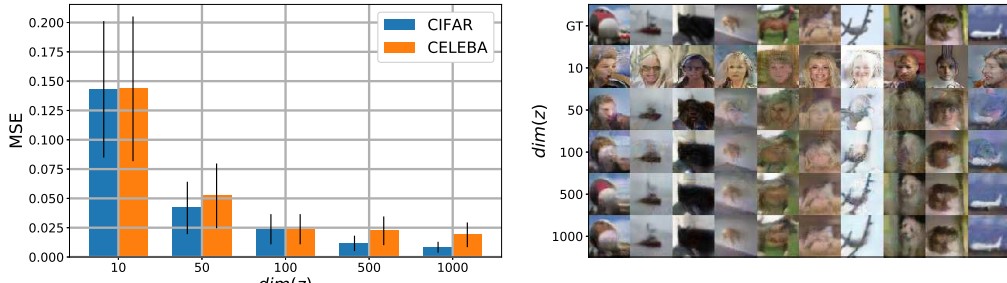

Figure 7: Left & Right: Reconstruction of CIFAR images in a WGAN-GP model trained on CelebA images. As the latent dimensionality is increased, the reconstruction is improved for both CIFAR-10 and CelebA. Right: With a small latent space, the face image features are retained, while with a large latent space, the CIFAR-10 images are faithfully reconstructed.

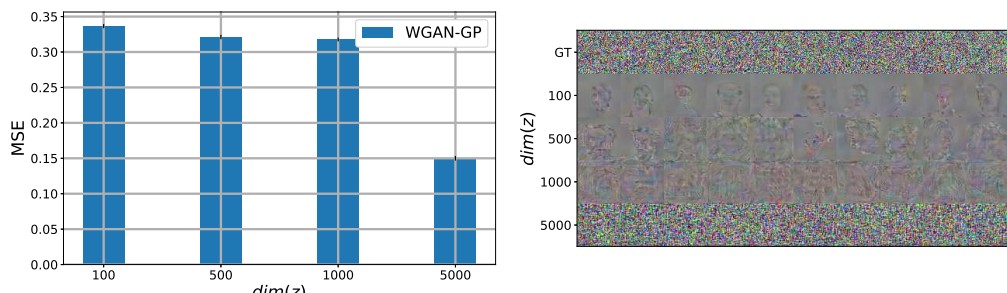

Figure 8: Reconstruction error of uniform noise images between $[-1, 1]$ using a generator network trained on CelebA images. With smaller latent spaces, it is still possible to see some key face features. A larger latent space manages to produce better reconstructions.

## 6 CONCLUSIONS

In this work, we show that by reducing the problem to a compression task, we can give a lower bound on the required capacity and latent space dimensionality of the generator network for the distribution estimation task. Relating these two different algorithms to each other, the literature surrounding AEs for invertability and dimensionality reduction, as well as the corresponding theoretical advancements are used. While in this work the encoder and the discriminator NNs use the same architecture, we have not discovered any relation between them. Still, the same architecture works well empirically for both task.

Using our framework we show various properties of generator networks. The perceptual image quality appears to be independent of the actual size of the latent space, which is in contrast to AE, where the visual quality improves if the dimensionality of the latent space is increased. However, the ability to reconstruct the training set correctly does depend on the initial latent space. Also the ability of the generator to reconstruct deviations from the original dataset, like a validation set or shifted images depends just as much on the initial latent space. However, the same cannot be said for reconstructing arbitrary noise images. Here the reconstruction ability is independent of the initial latent space unless it is chosen very large, suggesting that the generator has learned realistic natural image features. Here for smaller latent spaces we can still observe face like features. Our hypothesis is that the implicit bias induced by the generator architecture lends itself to generating natural images and GANs are skilled at that by learning primitives which can be combined to construct arbitrary images. In future works we want to use our setup to search towards better and more reliable generators for images.

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

ADDITIONAL MATERIAL

ARCHITECTURE AND HYPERPARAMTER SETTINGS

For our experiments a standard CNN was used as is described in Table 1. All the convolutions use $5 \times 5$ kernels. For the CIFAR-10 experiments we removed one of the up and downlayers and started with $1024$ feature maps. We used the ADAM optimizer with learning rate $1E - 4$ and $\beta_1 = 0.$ and $\beta_2 = 0.9$ similar to the original WGAN-GP as described by Gulrajani et al. (2017). We trained for $80.000$ generator iterations and 5 discriminator iterations per generator update for all of our networks. We also trained the AE for $80.000$ iterations with ADAM using the default hyperparameters. The inversion of the generator networks was done using $10.000$ ADAM steps using default hyperparameters. We found that the changes afterwards were negligible.

| Generator Architecture (h, w, $f_{out}$) |
| --- |
| Dense/BN/ReLU (to $(4, 4, 2048)$) |
| ResizeNN(to $(8, 8, 2048)$) |
| Conv2D/BN/ReLU (to $(8, 8, 1024)$) |
| ResizeNN(to $(16, 16, 1024)$) |
| Conv2D/BN/ReLU (to $(16, 16, 512)$) |
| ResizeNN(to $(32, 32, 512)$) |
| Conv2D/BN/ReLU (to $(32, 32, 256)$) |
| ResizeNN(to $(64, 64, 256)$) |
| Conv2D/BN/ReLU (to $(64, 64, 128)$) |
| Conv2D/TanH (to $(64, 64, 3)$) |

| Discriminator Architecture (h, w, $f_{out}$) |
| --- |
| SConv2D/LReLU(to $(32, 32, 256)$) |
| Conv2D/LReLU (to $(32, 32, 256)$) |
| SConv2D/LReLU(to $(16, 16, 512)$) |
| Conv2D/LReLU (to $(16, 16, 512)$) |
| SConv2D/LReLU(to $(8, 8, 1024)$) |
| Conv2D/LReLU (to $(8, 8, 1024)$) |
| SConv2D/LReLU(to $(4, 4, 2048)$) |
| Conv2D/LReLU (to $(4, 4, 2048)$) |
| Dense (to 1) |

Table 1: Architecture of the Discriminator and Generator networks. Initialization is done using Glorot & Bengio (2010)

PROOF OF CONCEPT ON MNIST

The initial experiments on MNIST validating the legitimacy of the approach are shown in Fig. 9. Instead of using a visual quality metric to determine if the resulting images are similar enough, we used the reconstructed training images of the GAN for training a LeNet network LeCun et al. (1998) to determine if the images are descriptive. The manifold of MNIST has been faithfully learned before by Shmelkov et al. (2018). In contrast to their work, we show that this only works given a large enough latent space and that the reconstruction error of the AE and the WGAN-GP converge for larger latent spaces.

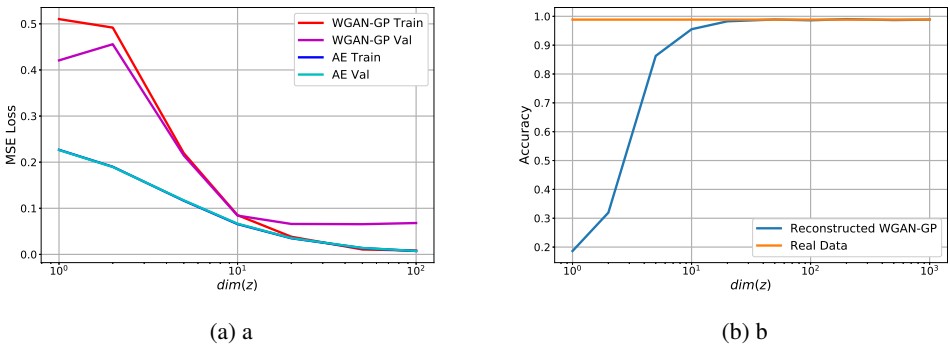

(a) a                                         (b) b

Figure 9: Mnist GAN and AE reconstructions in (a). Visual quality metrics are not really applicable on MNIST data so in (b) the reconstructed GAN images are used to train a LeNet to classify on MNIST and is compared to training on the true training set.

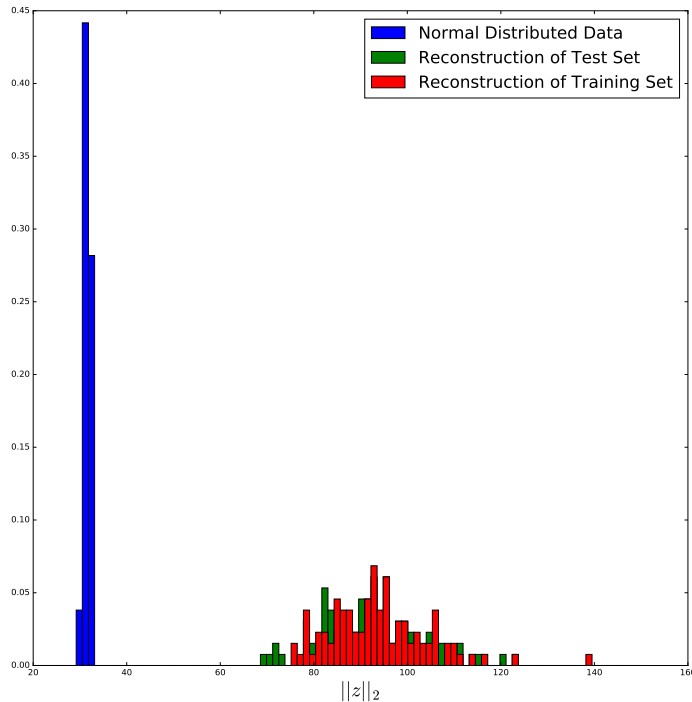

Figure 10: The distribution of the reconstructions without projections.

### DISTRIBUTION OF THE TRAINING SET IN THE GENERATOR MANIFOLD

In this section the main question is if the found reconstructions likely to appear in the generative model. For this reason the inverse optimization procedure is run without the projection onto the feasible set for $dim(z) = 1000$ on CIFAR-10. The resulting latent vector norm of the reconstructions is shown in Fig 10. The corresponding set of vectors, which is likely to appear in the training is shown in blue, while the training set reconstructions are shown in red and the test set reconstructions are shown in green. As is apparent therein, the overlap is virtually non-existant and this is the reason why we used a projected gradient method.

### ADDITIONAL GAN VS AE EXPERIMENTS

We also investigated the difference between the ability of GAN to reconstruct images of CIFAR-10 while trained on CelebA compared to AE. The results are shown in Fig. 11 and as is already shown in Fig. 7 the GAN algorithm is better at reconstructing CIFAR-10 images than CelebA images. However, this is not the case for the AE.

And as a final experiment we show visual results on CIFAR-10 images produced by an inverted GAN and by an AE in Fig. 12. As in the main paper we can the same behavior again. The AE produces initially blurry results and then sharp ones and the GAN produces sharp results of unrelated images and then produces images which are closer to the actual images.

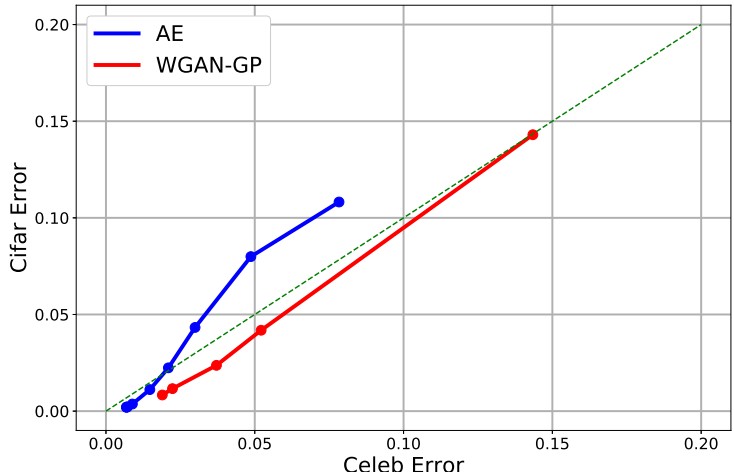

Figure 11: ROC Curve of training on CelebA and testing the reconstruction error of the NN on CIFAR-10 and CelebA. The desired behavior would be in the top left corner. The points correspond to a setting with a specific latent space. As the latent space decreases the error goes down. The points correspond to the target latent space size as is shown in Fig. 7

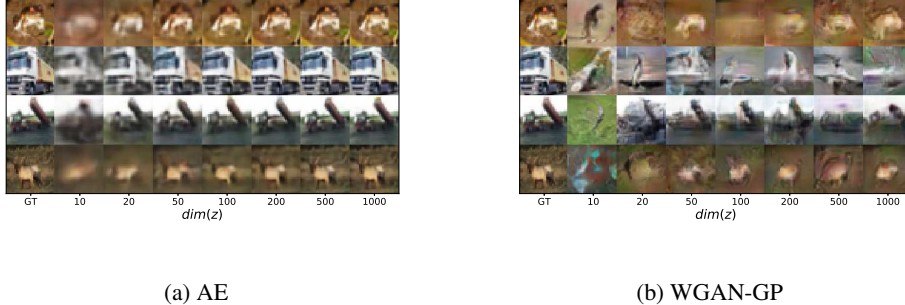

(a) AE                                          (b) WGAN-GP

Figure 12: The first column are the actual images. As we increase the latent space of the autoencoder we can see the image quality increasing. The same cannot be said about the GAN reconstructions. Those are of high quality even for low dimensional space, but correspond to other people.

