# OpenReview forum: "Impact of the latent space on the ability of GANs to fit the distribution"
_ICLR.cc/2020/Conference — Reject_

### Official Review · AnonReviewer1 · 2019-10-21
**Official Blind Review #1**

**Rating:** 1

**Review:**

Impact of the latent space on the ability of GANs to fit the distribution

This paper purports to study the behavior of latent variable generative models by examining how the dimensionality of the latent space affects the ability of said models to reconstruct samples from the dataset. The authors perform experiments with deterministic autoencoders and WGAN-GP on CIFAR, CelebA, and random noise, and measure MSE and FID as a function of the dimensionality of z.

My take: This paper does not offer any especially intriguing insights, and many of the conclusions the authors draw are, in my opinion, not supported by their experiments. The paper is confusingly written and hard to follow—throughout my read I struggled to determine what the authors meant, and it was not clear to me what this paper is supposed to contribute. The potential impact of this paper is very low, and I argue strongly in favor of rejection.

Notes:

My most critical complaint is the central experiment set of the paper: measuring MSE and FID as a function of Dim-Z for two models. First of all, the authors assume that the reduction in MSE as a function of dim Z is indicative of increased memorization in the GAN models. I disagree that this is the case; since the GAN-based reconstruction is done via optimization it is unsurprising that increasing dim Z increases the reconstruction quality, as you are literally giving the model more degrees of freedom with which to fit the individual samples. This is glaringly evident in Figure 8, where increasing dim Z renders the model better able to reconstruct a sample of pure noise, which is almost certainly not in its normal output distribution, (or if it is, is in there with staggeringly low probability). The fact that the higher dim-z models are better able to reconstruct the noise supports the notion that it is merely the number of degrees of freedom that matter in these experiments, rather than what the model actually learns.

Second, it is important to note that FID can be easily gamed by memorization, and for an autoencoder (which has direct access to samples) with an increasingly large bottleneck it is unsurprising that increasing dim-Z tends to decrease the FID, and equally unsurprising that increasing the dim-Z for the GAN does not tend to improve results, since this does not really allow the model increased memorization capacity (not to mention the relationship between performance and dim-Z has been explored before in GAN papers).

Third, the organization of the experimental section makes it very difficult to infer what the authors are trying to conclude from these experiments. The noise experiment is presented, but no insights or conclusions are drawn, other than (a) noting that the model has a harder time reconstructing the noise than training samples and (b) that lower dim models have a harder time reconstructing the noise, both of which are just restatements of the information presented in the figure rather than an actual insight or conclusion.

-I’m not really sure what the experiment in section 3 is supposed to show. This experiment is poorly described and lacking details. First of all, what is the loss function used there? Is this the output of the discriminator or the MSE between the output and a target sample? How is z* found and what does it represent—is it just a randomly sampled latent, or is it the latent that corresponds to the z-value which minimizes some MSE loss for a target sample? If it’s the latter, why is this notation not introduced until section 4? If it’s a latent, why are you calling it a data point? Why are there no axes and no scales on these plots? How is it clear that there is an optimization path from z0 to z*; is that supposed to be inferred from z0 having a higher value than z* or appearing to be directly uphill from z*, because it’s not clear to me that that is the case in Figure 2a. In general I did not find this experiment to support the conclusions the authors draw.

-Figure 4: It is important to note that FID can be trivially gamed by memorizing the dataset, and an autoencoder is much more well-suited to memorizing the dataset as it has direct access to samples (whereas a GAN must get them through the filter of the discriminator). Authors should test interpolation or hold-out likelihood for the  autoencoder, these models are not directly comparable in this manner.

-The presentation of this paper is, in general, all over the place. The authors should focus on writing such that each point follows the next, building progressively towards their results and insights, and making it easy for a reader to follow their train of thought.

“In this work, we show that by reducing the problem to a compression task, we can give a lower bound on the required capacity and latent space dimensionality of the generator network for the distribution estimation task.” At what point is this lower bound (either in terms of model capacity or latent space dimensionality) specified in the paper? Is Figure 3 supposed to be this lower bound, because to me it only indicates that the autoencoder tends to have a lower MSE, not that it conclusively lower bounds the memorization capacity of the GAN. Wouldn't a method like GLO which directly optimizes for memorization be a better lower bound for this, anyhow?

“We rely on the assumption, that less capacity is needed to reconstruct the training set, than to reconstruct the entire distribution” What does this phrase mean? Are the authors referring to the entire distribution of natural images, of which the training set is assumed to be a subset? Or do they mean the output distribution of the generator? This was not clear to me.


Minor:

-“ style transfer by Karras et al. (2018),”, and “anomaly detection (Shocher 2018).” StyleGAN is not a style transfer paper, and InGAN  is not about anomaly detection. Please do not incorrectly summarize papers.

-“Trained GAN newtworks” While amusing, this is a typo. Please thoroughly read your paper and correct all typos and grammatical mistakes, like “combiniation.”

-“…that an accurate reconstruction of the generator manifold is possible works using first order methods”  The word “works” seems to be out of place here. Again, please thoroughly proofread your paper.

-The legend in Figure 2 has a white background, making the white x corresponding to z0 invisible. Please fix this, and add appropriate axes to this plot.

-Figure 7 and 8 may in fact have error bars, but they are not described (are they 1 std or another interval?) or referenced, and in Figure 8 (if these are error bars) they are nearly invisible.



**Experience Assessment:**

I have published in this field for several years.

**Review Assessment: Checking Correctness Of Derivations And Theory:**

I carefully checked the derivations and theory.

**Review Assessment: Checking Correctness Of Experiments:**

I carefully checked the experiments.

**Review Assessment: Thoroughness In Paper Reading:**

I read the paper thoroughly.

---

### Official Review · AnonReviewer2 · 2019-10-22
**Official Blind Review #2**

**Rating:** 1

**Review:**

The work performs a systematic empirical study of how the latent space design in a GAN impacts the generated distribution. The paper is well-written and was easy to read. Also, I find this to be an interesting and promising direction of research.

The convergence proof in Goodfellow (2014) assumes that the data distribution has a density, and essentially states that the JS-divergence is zero if and only if the two distributions are the same. In practice, the data distribution is discrete, while the latent distribution has a probability density function. It is not possible to transform a density into a discrete distribution by a continuous map and neural networks are always continuous by construction. In theory, as training progresses, more and more latent mass will be pushed on the discrete samples and no minimizer exists (unless the function space of generator is constrained or the real distribution is smoothed out a bit).

Since it is not possible to assess whether the GAN training has converged due the nonconvexity of the energy and non-existence of a global optimizer, the empirically observed results might be very specific to the chosen optimization procedure, stopping criterion, dataset, hyper-parameters, initialization, network architectures, etc etc.  It is a challenge to study the choice of latent space in a somewhat "isolated" way. These issues should be discussed in the paper and the reader should be made aware of such problems.

Another point, could it be, that by increasing the dimension of the latent space, one makes it easier for the nonconvex optimization in (5) to converge to "unlikely but realistic looking samples"? I think this is not too far-fetched, as increasing the dimension of an optimization problem often makes local optimization less likely to get stuck at local optima. Also it might not be the best idea to optimize (5) with Adam since it is not a stochastic optimization problem and there are provably convergent solvers out there for this problem class.

Since it is possible to evaluate the likelihood of the optimized reconstructions that are nearby the data points, one could check whether this is indeed the case. While constrained not to be too unlikely, I wonder whether the likelihood increases or decreases with the dimensionality of the latent space and this would make an interesting plot.

Unfortunately, I did not understand the connections to auto-encoders, as they might optimize a fundamentally different criterion than GANs. In particular "In principle, getting from an AE to a GAN is just a rearrangement of the NNs. " is unclear to me.

Also, what is meant by lower-bound? Is the claim that the reconstruction error in an auto encoder will be lower, than if one optimizes the latent code in a GAN to reconstruct the input? Figure 3 seems to support this hypothesis, but I don't have an intuition why this should be true and have some doubts. A mathematical proof seems out of reach.

I have trouble to understand the "intuition that the AE complexity lower-bounds the GAN complexity." Before reading this paper, my intuition was the opposite: If the generator distribution covers the real distribution, the reconstruction error for GAN is zero. Intuitively, it seems a much easier task to somehow cover a distribution than to minimize an average reconstruction error.

The connection of WGANs to the L2 reconstruction loss in the auto-encoder is very hand-wavy. It is still an open question whether WGANs actually have anything to do with the Wasserstein distance. People working in optimal transport doubt this, due to huge amount of approximations going on.

At this point I'm reluctant to recommend acceptance, as the paper tries to connect things, which for me are quite disconnected and the evaluations of reconstruction error, etc. might depend in intricate ways on the nonconvex optimization procedures.

Minor suggestions, typos, etc (no influence on my rating):

- What is the "generated manifold" that is talked about in the introduction, contributions and throughout the paper? To me, it is not directly clear that the support of the transformed distribution will be a manifold (especially if G is non-injective). Anyway, the manifold structure is nowhere exploited in the paper, so I suggest to call it "transformed latent distribution".

- Had to pause a little bit to understand Eq. 2 (simple polynomial interpolation). It is unnecessary to show the explicit form, as I'm sure no one doubts the existence of a smooth curve interpolating a finite set of points in R^d.

- Equations should always include punctuation marks.

- Eq. 5: dim --> \text{dim} and s.t. --> \text{s. t.}

- Fig 3b: the red curve is missing or hidden behind another curve.


**Experience Assessment:**

I have published one or two papers in this area.

**Review Assessment: Checking Correctness Of Derivations And Theory:**

N/A

**Review Assessment: Checking Correctness Of Experiments:**

I assessed the sensibility of the experiments.

**Review Assessment: Thoroughness In Paper Reading:**

I read the paper at least twice and used my best judgement in assessing the paper.

---

### Official Review · AnonReviewer3 · 2019-10-28
**Official Blind Review #3**

**Rating:** 1

**Review:**

Summary: The paper explores the influence of the dimensionality of the latent space to the quality of the learned distributions for autoencoders (AE) and GANs (more precisely Wasserstein GANs). In particular, the paper looks at the ability of the learned AE or GAN to reconstruct the training images, the visual quality of the images as the dimension of the latent space increases, and the ability to reconstruct images not in the training set (structured in some ways).

Evaluation: While the general flavor of question the paper studies is undoubtedly interesting, I found the paper severely lacking both in terms of the quality of writing (in particular, I was at confused about the goal of various sections/experiments), as well as the significance of the results the authors observe (and how they are reported -- I found them to be oversold).

Regarding the quality of results:

* The paper primarily talks about the ability of AEs and GANs to *reconstruct* images, either in the training set, or in the test set, or in some different dataset altogether (e.g. shifted images, different image dataset). This is a problematic thing on multiple levels: first, the goal of a GAN or AE is to fit a distribution -- merely having a data point in its domain says nothing about the *probability* of that point; second, the way these "spans" are tested is via running a gradient descent search for a pre-image for the data point. The authors never comment or explore whether the problem may *not* be that these data points are not in the image of the GAN, but rather that the optimization procedure doesn't succeed. (And indeed, increasing the dimensionality of the latent space may act as "overparametrization" for this gradient descent procedure, making it more likely to succeed.)

Finally, there are some fairly arbitrary choices in the entire experimental setup: why WGANs vs another architecture -- are the GAN results sensitive to architecture choices? why AEs and not VAEs (and with what variational posterior) -- how sensitive are the observations here to choosing the most vanilla variant of autoencoders? These are all questions that invariably linger after reading the paper.

Regarding the quality of writing:

* There are various sloppy sentences in crucial parts of the paper. I will only list a few:
-- "Once a suitable AE for a dataset is found, the decoder part of is used as the generative model of choice." -- this seems to suggest a semi-synthetic setup where an AE is trained to use as a generator of a data set for which a GAN is fit. I never saw this setup in Section 5 -- although this would be a good way to test "relative" representational power of GANs and AEs.
-- "In principle, getting from an AE to a GAN is just a rearrangement of the NNs" in Section 5.1 -- I wasn't sure what this is supposed to mean, and this is a critical part of that section.
-- "The AE network acts as a lower bound for the GAN algorithm, therefore validating our intuition that the AE complexity lower-bounds the GAN" in Section 5.1 -- also very sloppy, and I'm not sure what it means -- I guess the authors mean the reconstruction performance of AE is a lower bound on the GAN reconstruction. Not sure what this has to do with "complexity".

* Various sections are meandering, and I wasn't sure what the goal is. Just a few examples: section 3 spends a lot of time talking about known theoretical results wrt. to invertibility of random-like neural nets. It wasn't clear to me how this relates to the results in Section 5, especially since the authors never leverage/talk about these theory results again. (Instead, they study empirical invertibility via gradient-descent based procedures.) Similarly, interpolating by polynomials is talked about in (2), seemingly without any point.


**Experience Assessment:**

I have published in this field for several years.

**Review Assessment: Checking Correctness Of Derivations And Theory:**

I carefully checked the derivations and theory.

**Review Assessment: Checking Correctness Of Experiments:**

I carefully checked the experiments.

**Review Assessment: Thoroughness In Paper Reading:**

I read the paper thoroughly.

---

### Decision · Program_Chairs · 2019-12-19

**Decision:**

Reject

**Comment:**

The reviewers have pointed out several major deficiencies of the paper, which the authors decided not to address.